# Cancer Cells Shuttle Extracellular Vesicles Containing Oncogenic Mutant p53 Proteins to the Tumor Microenvironment

**DOI:** 10.3390/cancers13122985

**Published:** 2021-06-15

**Authors:** Bibek Bhatta, Ishai Luz, Christian Krueger, Fanny Xueting Teo, David P. Lane, Kanaga Sabapathy, Tomer Cooks

**Affiliations:** 1The Shraga Segal Department of Microbiology, Immunology and Genetics, Ben-Gurion University of the Negev, Beer-Sheva 84105, Israel; bhatta@post.bgu.ac.il (B.B.); ishail@bgu.ac.il (I.L.); 2Division of Cellular & Molecular Research, Humphrey Oei Institute of Cancer Research, National Cancer Centre Singapore, Singapore 169610, Singapore; christian.heiko.krueger@nccs.com.sg (C.K.); fannyteo.xt@gmail.com (F.X.T.); cmrksb@nccs.com.sg (K.S.); 3p53 Laboratory (p53Lab), Agency for Science, Technology, and Research (A*STAR), Singapore 138648, Singapore; dplane@p53lab.a-star.edu.sg; 4Cancer and Stem Cell Biology Program, Duke-NUS Medical School, Singapore 169857, Singapore

**Keywords:** mutant p53 protein, extracellular vesicles, tumor microenvironment

## Abstract

**Simple Summary:**

In addition to the classical cell-to-cell communication patterns, extracellular vesicles (EVs) are instrumental in conveying molecular messages across cell types and have the potential to mediate changes at a tissue level. Since it is now appreciated that carcinomas are fundamentally reliant on two-way communication with activated cells in the tumor microenvironment, elucidating the roles of EVs exchange and of the cargo that is transferred is essential to obtain a thorough understanding of tumor progression. This study reveals that mutant p53 proteins—the result of the most frequent mutated gene in human cancer—are packed into EVs and delivered to neighboring cells with the potential to reprogram immune cells and subsequently establish a positive feedback loop that will enhance tumor progression. This non-cell autonomous role of mutant p53 is evidence of an extra layer of communication that is orchestrated by smaller vesicles that transfer oncogenic elements between cellular entities. Building on the foundation of our work on mutant p53, future studies may aim to characterize the potential activation of additional oncogenes, thus opening new paths of research at the interface of extracellular vesicles, cancer, and evolution.

**Abstract:**

Extracellular vesicles (EVs) shed by cancer cells play a major role in mediating the transfer of molecular information by reprogramming the tumor microenvironment (TME). *TP53* (encoding the p53 protein) is the most mutated gene across many cancer types. Mutations in *TP53* not only result in the loss of its tumor-suppressive properties but also results in the acquisition of novel gain-of-functions (GOF) that promote the growth of cancer cells. Here, we demonstrate that GOF mutant p53 proteins can be transferred via EVs to neighboring cancer cells and to macrophages, thus modulating them to release tumor supportive cytokines. Our data from pancreatic, lung, and colon carcinoma cell lines demonstrate that the mutant p53 protein can be selectively sorted into EVs. More specifically, mutant p53 proteins in EVs can be taken up by neighboring cells and mutant p53 expression is found in non-tumor cells in both human cancers and in non-human tissues in human xenografts. Our findings shed light on the intricate methods in which specific GOF p53 mutants can promote oncogenic mechanisms by reprogramming and then recruiting non-cancerous elements for tumor progression.

## 1. Introduction 

Malignant transformation is driven by an accumulation of genetic mutations. When the balance between oncogenic and tumor suppressing events is skewed towards the former during the transformation process, growth control is compromised, which could result in cancer. Beyond these well established cell-autonomous driving forces, the entire process is also regulated and controlled by myriad factors derived from the non-tumoral surrounding tissue that constitutes the tumor microenvironment (TME) [1,2,3]. While many characteristics of TME modulation are well established, the biochemical basis of TME conversion is not well understood. In this respect, it is becoming increasingly clear that extracellular vesicles (EVs) play a major role in shuttling bioactive molecules including proteins, DNA, mRNA, and non-coding RNAs from one cell to another, which mediates the transfer of molecular information and the reprogramming of recipient cells [4,5,6]. Tumor cells release excessive amount of EVs, which may affect tumor initiation, growth, progression, metastasis, and drug resistance [7,8]. However, the question of whether EVs play an important role in TME subversion via their transfer of mutated protein products from the cancer cells is sparsely studied. 

Mutations in TP53 (encoding for the p53 protein), which is the master tumor suppressor, are the most common molecular event in human cancers: TP53 is mutated or inactivated in over 50% of all human cancers [9,10]. In certain histotypes, such as serous ovarian cancer, the rate of TP53 mutations can even exceed 95% [11]. 

While mutations in TP53, which are often missense mutations, result in the abrogation of wild-type (WT) p53-mediated tumor suppressive functions, some mutations have been reported to endow mutant p53 proteins with novel oncogenic properties termed gain-of-function (GOF) activities [12,13,14,15]. These GOF activities dramatically alter tumor cell characteristics primarily by their interactions with other cellular proteins and their regulation of cancer cell transcriptional programs [16,17,18,19,20,21]. However, studies demonstrating mutant p53 GOF have predominantly focused on cell-autonomous mechanisms (namely, events that affect the cancer cell itself). Nevertheless, the direct cross talk between cells harboring mutant p53 and their microenvironment, particularly via EVs, is gaining increasing attention. We previously showed that EVs shed by mutant p53 colon cancer cells could reprogram neighboring macrophages by rendering them tumor-supportive [22]. Since then, additional evidence accumulated has indicated that EVs from cancer cells harboring mutant p53 can mediate intercellular communication in the TME resulting in the promotion of tumor progression and metastasis [23,24]. 

Although mutations in p53 are found mainly in cancer cells, sporadic reports have described the presence of mutant p53 in normal untransformed cells. For example, untransformed hepatocytes have been shown to harbor the p53 mutant R249S, which is a typical mutation found in hepatocellular carcinomas [25]. Similarly, p53 mutations were noted in the stromal compartment of sporadic breast cancers and were correlated with nodal metastasis [26]. In this study, immunohistochemical analyses revealed levels of protein accumulation that strongly support the presence of mutant p53 in the stromal cells. However, these findings were not validated since the sequencing of micro-dissected samples revealed that they did not carry any p53 mutations [27]. Since these initial works were performed using generic reagents to detect mutant p53 that could not discriminate between the mutant and the WT p53, the validity of the resultant data has always been questionable. To that end, highly specific monoclonal antibodies were generated against three of the hot-spot p53 mutations. These antibodies do not cross-react with WT p53 or with other p53 mutants [28]. We therefore hypothesized that although mutant p53 proteins could indeed be found in TME cells, these proteins are exogenously sourced and shuttled into the untransformed cells via EVs. Notably, a recent study has demonstrated that mutant p53 proteins are found in EVs and could, by secretion, convert neighboring fibroblasts into cancer-associated fibroblast (CAFs) [29]. Supported by these findings, we tested the effect of GOF p53 mutants shuttled through EVs to neighboring cancer cells that lack mutant p53 and macrophages (a key member of TME). In this report, we corroborate the fundamental hypothesis suggesting the cancer cells that harbor GOF p53 mutants can package these mutant proteins in EVs and deliver them to neighboring cancer cells and macrophages. In particular, mutant p53s harboring EVs were able to educate macrophages to upregulate and increase the secretion of tumor-supporting cytokines. 

## 2. Materials and Methods

### 2.1. Cell Culture

All cell lines were grown and maintained as per ATCC guidelines. In all EVs experiments, media underwent an overnight ultracentrifugation to eliminate bovine EVs. In general, PANC-1 cells (a gift from the group of Professor Moshe Oren, Weizmann Institute of Science, Israel) were grown in Dulbecco’s Modified Eagle Medium (DMEM) (Gibco (Thermo-Fisher), Waltham, MA, USA) and HCT116 (from the group of Bert Vogelstein, Johns Hopkins Hospital, USA) were maintained in McCoy medium (Gibco (Thermo-Fisher), Waltham, MA, USA). H358 cells (from the group of Curtis C. Harris, NCI, NIH, USA) were cultured in Roswell Park Memorial Institute (RPMI) 1640 medium (Gibco, (Thermo-Fisher), Waltham, MA, USA) with 1% l-glutamine (Gibco, (Thermo-Fisher), Waltham, MA, USA). The amount of 10% fetal bovine serum (FBS) (Gibco (Thermo-Fisher), Waltham, MA, USA) and 1% Penicillin (100 U/mL) and streptomycin (100 U/mL) (Gibco (Thermo-Fisher), Waltham, MA, USA) were included in all media. Cells were grown at 37 °C supplemented with 5% CO_2_. All cell lines were diluted twice a week and used until passage 15. 

Mouse tail-tip fibroblasts expressing wild-type (WT) p53, heterozygous for p53 (+/−) and heterozygous for mutant R172H p53 (172/+) were treated without (Ctrl) or with 10 Gy g-irradiation (IR) and harvested 2 h later and used for immunoblot analysis.

### 2.2. Macropahge Generation and Culture

THP-1 (a gift from the group of Dr. Neta Regev-Rudzki, Weizmann Institute of Science, Israel) were cultured in Roswell Park Memorial Institute (RPMI) 1640 medium with 1% l-glutamine, 10% fetal bovine serum (FBS), 1% Penicillin (100 U/mL) and streptomycin (100 U/mL) and β-mercaptoethanol (0.05 mM) (Gibco, Waltham, MA, USA). Differentiation of THP-1 monocytes into macrophages were achieved via PMA (100 ng/mL) (Sigma Aldrich, St. Louis, MO, USA) for 24 h and were allowed to recover for an additional 24 h. Three doses of 20 µg of EVs were added for macrophage reprogramming experiments. After two days post-addition of EVs, medium was collected, RNA was extracted (Qiagen, Hilden, Germany), and transcribed to cDNA (qscript cDNA synthesis kit). cDNA was exposed to primers for qRT-PCR, as shown in Table 1. The medium was subjected to hTNFα ELISA as per manufacturer’s guidelines (DuoSet ELISA development system, Cat number: DY210-05, R&D Systems, Minneapolis, MN, USA).

### 2.3. Isolation of Exosomes/EVs

A standard serial ultra-centrifugation method was utilized for the isolation of EVs. Briefly, cells were grown in a 150 mm cell culture plate (Biofil, Guangzhou, China) to reach 80% confluence. Unless otherwise mentioned, 100 mL of medium was collected and centrifuged at 500× *g* for 5 min followed by 10,000× *g* for 30 min and then passed through 0.22 µM filters to remove cell contaminants and debris. The medium was then centrifuged at 100,000× *g* for 90 min in 4 °C. The pellet obtained was resuspended in PBS 1X and then subjected to another round of 100,000× *g* for 90 min in 4 °C centrifugation. The supernatant was discarded and the pellet was resuspended in PBS 1X. When indicated, the EVs isolated by ultra-centrifugation was further laid in the discontinuous gradient of Iodixanol (Optiprep)—sucrose (40%/20%/10%/5%)—and centrifuged overnight at 27,000 RPM in 4 °C to achieve pure fractions of EVs. Fractions 6 to 15 were collected and centrifuged for an additional 90 min at 100,000× *g* in 4 °C. For size exclusion chromatography (SEC), EVs isolated from ultra-centrifugation were passed through the 70 nm column (qEV, Izon science) according to the manufacturer’s guidelines. 

### 2.4. Nanoparticle Tracking Analysis

Nanoparticle tracking analysis measurements were performed using a NanoSight NS500 Instrument (NanoSight NTA 2.3) (Salisbury, UK) following the manufacturer’s instructions. Samples were processed in duplicates and diluted with PBS (1:100) before analysis. NTA post-acquisition settings were optimized and kept constant between samples. Three videos 60 s long were recorded per sample and were analyzed to give the mean, mode, and median particle sizes together with an estimate of the number of particles.

### 2.5. Transmission Electron Microscopy

EVs were isolated from pancreatic cancer cell line harboring mutp53 (PANC-1) utilizing ultracentrifugation followed by SEC. FEI Tecnai T12 G2 TWIN transmission electron microscope (Hillsboro, OR, USA) operating at 120 kV was used. The images were taken with Gatan 794 MultiScan CCD camera. In short, the samples were prepared in the following manner: 2.5 µL of the sample was applied on to 300 mesh copper grid and the excess liquid was blotted with filter paper after 1 min. The grid was dried in air for 1 min, followed by applying 5 µL of uranyl acetate 2% for negative staining in order to increase the sample contrast. Next, the grid was blotted once more in order to remove the excess uranyl acetate. Finally, the grid was dried in air before insertion in to the microscope.

### 2.6. Confocal Microscopy

PANC-1 p53 K/O cells or THP-1-derived macrophages were seeded on glass bottom chambers (ibidi, Grafelfing, Germany). EVs isolated from PANC-1 scramble cells were labelled with PKH26 dye (Sigma Aldrich, St. Louis, MO, USA) (1 µM), as per manufacturer’s guidelines. Briefly, 1 µM PKH26 was exposed to EVs followed by iodixanol gradient differentiation of fragments. Fragments 6 to 15 were combined, centrifuged at 100,000× *g* for 1.5 h in 4DC, and 20 µL of labelled EVs were added to each chamber for 18 h. As a control, cells were also treated with 20 µL of 1 µM dye-only solution (without EVs). The cells were washed and viewed with oil immersion under 63X lens in LSM880 Zeiss Microscope (Zeiss, Jena, Germany).

### 2.7. Flow Cytometry

EVs from PANC-1 cells (CAS9 scramble and CAS9 p53 KO) were either stained with PKH26 (as mentioned above) or DIR dye. Briefly, DiR’ DiIC18(7) (1,1′-Dioctadecyl-3,3,3′,3′-Tetramethylindotricarbocyanine Iodide) from Thermo Fisher (Cat number: D12731, Waltham, MA, USA) was used in a concentration of 2 µM in 1000 µL of PBS1X with EVs or without EVs (dye control). EVs were incubated with the dye for 30 min at 37 °C followed by centrifugation for 1.5 h at 100,000× *g*. The stained EVs were suspended in 100 µL of PBS 1X. The recipient cells (PANC-1 CAS9 scramble or CAS9 p53 KO) were seeded in a 96 well plate at a concentration of 60,000 cells/well and exposed to 10 µg of EVs for 16 h. The occurrence 15,000 events were noted for each well using PE-A filter and APC-A750-A filter in cytoflex. Data analysis was conducted using the cytexpert software. 

### 2.8. Antibodies

Mouse Monoclonal anti-p53 (DO-1) was purchased from Santa Cruz Biotechnology, Santa Cruz, CA, USA. Rabbit anti-CD9 and HRP-goat anti-rabbit were purchased from System Bioscience, Palo Alto, CA, USA. Rabbit Monoclonal anti-calnexin (C5C9), IC12, and mouse anti-alix were purchased from Cell Signaling, Danvers, MA, USA. SRSF10, HRP anti-mouse, and Mouse anti-GAPDH were purchased from Sigma-Aldrich, St. Louis, MO, USA. Anti-ITGB4 was purchased from Cell Signaling Technology, Danvers, MA, USA. 

### 2.9. Immunoblotting/Western Blot

EVs pellet obtained from ultra-centrifugation and/or from density gradient was lysed with 30 µL of RIPA 1X buffer supplemented with Protease inhibitor cocktail (1:100, Thermo Fisher Scientific, Waltham, MA, USA). Protein concentration in the lysates were determined via the Pierce BCA protein assay kit (Thermo Fisher Scientific, Waltham, MA, USA). Lysates were mixed with sample buffer (5X) (Thermo Fisher Scientific, Waltham, MA, USA) and boiled at 95 °C for 10 min, loaded into gel (Invitrogen Novex WedgeWell 4 to 20%, Tris-Glycine, 1.0 mm, Mini Protein Gel, 10-well, Thermo Fisher Scientific, Waltham, MA, USA), and separated via PAGE. Proteins were then transferred into the nitrocellulose membrane (Greiner, 10-6000-02), blocked for 1 h at room temperature with 5% BSA in TBS and then followed by exposure to primary antibodies (dil 1:1000) overnight at 4 °C: anti-p53, anti-calnexin, anti-CD9, anti-Alix, anti-GAPDH, and anti-ITGB4. The membranes were then washed and incubated with HRP-conjugated secondary antibody (dil 1:5000) for 1 h at room temperature, washed, and exposed to SuperSignal West Pico PLUS Chemiluminescent Substrate (Thermo Fisher Scientific, Waltham, MA, USA) as per manufacturer’s protocol before visualizing the membranes in iBrightCL1000 (Invitrogen, A32749, Carlsbad, CA, USA). Lysates from mouse tail-tips fibroblasts were prepared as described in [30]. IC12 is a pan-p53 antibody that detects all forms of p53 and was used at a 1:1000 dilution. The R175H antibody was used to detect the R172H mutation (human R175H) at a 1:3000 dilution [28].

### 2.10. Exosome Shaving

We used the XPEP kit (Systems Biology, Cat. #XPEP100A-1) according to manufacturer’s instructions. Briefly, we added the shaving buffer to the EVs pellet and allowed an enzymatic digestion of free proteins and surface proteins. After reduction, alkylation, and digestion the compartment of EV proteins, which were protected by the EV membranes, were run by a WB (as described above) and the presence of mutant p53 was tested using the DO-1 antibody. 

### 2.11. Xenograft Tumor Generation

Colo 320DM cell line-derived xenografts were generated by injecting a mixture of 5 × 10^5^ cells in 50 µL PBS and 50 µL of Matrigel into the flanks of 8 weeks old C.B.17 SCID mice. When tumors grew to 500 mm^3^, they were excised (including skin and adjacent mouse tissue), fixed in 10% formalin overnight, washed in PBS once, and preserved in 70% ethanol until paraffin embedding for histological analysis. Deparaffinized five-micron sections were used for immuohistochemical analysis.

### 2.12. Human Tumor Samples

The study that included human tissues was approved by the Institutional Review Board at the NIH and all individuals participating in the NCI-MD case-control study signed informed consents for the collection of biospecimens, personal, and medical information. Colorectal cancer tumors were resected at the University of Maryland, Baltimore (UMB) School of Medicine, Baltimore, MD, USA. The IRB Federal Wide Assurance (FWA) number is FWA00007145. 

### 2.13. Immunohistochemistry 

Immunohistochemical staining was performed using a primary antibody to human p53 (DO-1, Abcam Ab1101) or human SRSF10 (HPA053831, Sigma-Aldrich) at a 1:300 or 1:150 dilution, respectively. An anti-mouse and anti-rabbit secondary antibody kit (K5007, Dako) was used in both cases. Signal development time was 2 and 6 min for anti-p53 and anti-SRSF10 antibodies, respectively. Slides were subsequently counter-stained with Haematoxylin, dehydrated, and covered in DPX Mounting Media as described [28]. Sections of human tumor tissues were obtained from the Lab of human carcinogenesis (NCI, Bethesda, MD, USA) and stained with the anti-p53 antibody as described. Images of stained slides were acquired using a Leica DM2000 microscope with a Leica DFC295 camera.

### 2.14. Mass-Spectrometry

The samples were dissolved in 10 mM DTT, 100 mM Tris, and 5% SDS. They were sonicated and boiled at 95 °C for 5 min and precipitated in 80% acetone. The protein pellets were dissolved in 9 M Urea and 400 mM ammonium bicarbonate, reduced with 3 mM DTT (60 °C for 30 min) modified with 10 mM iodoacetamide in 100 mM ammonium bicarbonate (room temperature 30 min in the dark), and digested in 2 M Urea and 25 mM ammonium bicarbonate with modified trypsin (Promega, Wisconsin, WI, USA) overnight at 37 °C in a 1:50 (M/M) enzyme-to-substrate ratio. The tryptic peptides were desalted using C18 tips (Top tip, Glygen) dried and re-suspended in 0.1% Formic acid.

The peptides were resolved by reverse-phase chromatography on 0.075 X 180-mm fused silica capillaries (J&W) packed with Reprosil reversed phase material (Dr Maisch GmbH, Ammerbuch, Germany). The peptides were eluted with linear 60 min gradient of 5 to 28%, 15 min gradient of 28 to 95%, and 25 min at 95% acetonitrile with 0.1% formic acid in water at flow rates of 0.15 μL/min. Mass spectrometry was performed by the Q Exactive HF mass spectrometer (Thermo) in a positive mode using repetitively full MS scan followed by collision induces dissociation (HCD) of the 18 most dominant ions selected from the first MS scan. 

The mass spectrometry data from three biological repeats were analyzed using the MaxQuant software 1.5.2.8 (Mathias Mann’s group) vs. the human proteome from the Uniprot database with 1% FDR (false discovery rate). The data were quantified by label free analysis using the same software. Statistical analysis of the identification and quantization results was performed using the Perseus 1.6.10.43 software (Mathias Mann’s group).

## 3. Results

### 3.1. GOF Mutant p53 Proteins Are Sorted into EVs

In order to investigate whether cancer cells harboring GOF mutant p53 can encapsulate the mutated proteins into EVs and release them using the exosomal machinery, we chose to test several complementary cellular systems that originate from several different types of cancer and that contain a series of different p53 GOF mutations. In PANC-1 pancreatic ductal adenocarcinoma cells, we knocked out the R273H endogenous mutant p53 using the CRISPR-Cas9 system (Figure 1A and Appendix A). In H358 lung carcinoma cells, we used Tetracycline-controlled transcriptional activation (Tet-On inducible) cell system overexpressing several different p53 GOF mutants (V157F, R175H, R249S, and R273H), which simulates inactivated p53 compared with the WT form and an empty vector (ConVec) (described in detail here [22]). In addition, we used the isogenic set of HCT116 colon carcinoma cells that was harboring either WT p53 (+/+), p53 Null (−/−), or mutant p53 (−/R248W) as published in [31]. EVs from these cells were isolated and subjected to several characterization steps that included nanoparticle tracking analysis (NTA) (Figure 1B), electron microscopy (Figure 1C), and immunoblotting (Exo-Check Exosome Antibody Arrays, System Bioscience) to detect different exosomal markers. NTA results (Figure 1B) were similar to what we previously showed in HCT-116 cells and H358 cells [22] and TP53 status did not dramatically affect the physical characteristics of PANC-1 EVs. The mean sizes and concentrations were 136.4 nm and 1.04 × 10^9^ particles/mL for the scrambled control and 134.1 nm and 4.57 × 10^8^ particles/mL for the p53 KO EVs. Electron microscopy revealed a typical cup-shaped morphology of EVs (Figure 1C). Notably, immunoblotting (Figure 1D) led to the detection of different exosomal markers such as TSG101, CD63, Annexin 5 associated with the EVs, and GM130, a golgi marker or cis-golgi matrix protein, which was considered to be a negative control in exosomes, was found to be absent in the EVs, indicating EVs to be free of cellular contaminants. Furthermore, in a mass-spectrometry proteomic assay, we analyzed proteins from PANC-1 EVs isolated either by standard ultracentrifugation (UC) in combination with a size-exclusion (SEC) step (IZON qEV column). Among several exosomal markers, CD9 was detected in both PANC-1 scrambled and p53 KO EVs (Figure 1E). We therefore focused on CD9 as our major exosomal marker in the following experiments.

After characterizing the EVs, we tested whether they contain mutant p53 proteins. In order to validate that the mutant p53 protein is inside the vesicles and not pulled down as a free protein, we used Iodixanol-sucrose density gradient fractionation to distinguish between EVs and particles and free bio-molecules. In PANC-1 cell line, mutant p53 was associated with CD9 detected in fractions 6–7 (Figure 2A and Appendix A). When comparing EVs released by HCT116 (+/+) and HCT116 (−/R248W) cells, we observed the sorting of the mutant but not the WT form of p53 in these EVs, which is again associated with the exosomal markers Alix and CD9 (Figure 2B and Appendix A). We also treated H358 cells with Doxycycline (Dox) to induce the expression of either WT, Null (empty vector control), or several GOF p53 mutants and then isolated EVs from all treated cells. Notably, the mutant but not the WT proteins were sorted into EVs (Figure 2C and Appendix A). To confirm that the EVs encapsulated the mutant p53 proteins inside them, we subjected the EVs isolated from H358 cells to “exosome shaving” protocols where only the proteins inside the vesicles’ membrane are protected from the proteolytic process. We used two exosomal markers: One representing the inner-vesicle population of proteins (Alix) and another representing the outer-vesicle population of proteins (ITBG4). As could be seen in Figure 2D and Appendix A, no mutant p53 proteins were affected by the exosome shaving. While Alix (the inner protein control) was also preserved, the outer-membrane part of ITBG4 was chopped by the shaving treatment. These results suggested that GOF mutant p53 proteins are carried inside the EVs and are specifically sorted into EVs since this phenomenon does not occur with the WT form of p53.

### 3.2. Vesicular Mutant p53 Proteins Are Taken Up by Neighboring Cancer Cells 

After determining that cancer cells harboring GOF mutant p53 can release EVs enriched with the mutant p53 proteins, we wanted to test whether these mutant-p53-containing-EVs are taken up by neighboring cancer cells. We labeled EVs from PANC-1 cells harboring mutant p53 (CAS9 scrambled) with the PKH26 lipophilic dye. The fluorescently labeled EVs were shown to be taken up by recipient PANC-1 cells that did not carry mutant p53 (CAS9-p53-KO) (Figure 3A). Additionally, we incubated PANC-1 CAS9-p53 KO cells with EVs harboring oncogenic mutant p53 proteins and, using the immune-fluorescence approach, we were able to see the perinuclear localization of oncogenic mutant p53 protein in the recipient cells (Figure 3B,C). The delivery of cargos by cancer-EVs to nuclear region of recipient cells is in line with other published reports [32,33]. To further endorse our results, we employed a direct approach where we incubated the PANC-1 CAS9-p53 KO cells with EVs isolated from PANC-1 harboring the mutant p53 (CAS9 scrambled) and performed immunoblot using specific antibody against p53. While after 12 h of incubation no mutant p53 was detected in the recipient cells (data not shown), after 24 h the mutant p53 protein was observed, which suggests an EV-based transfer between the cells (Figure 3D and Appendix A). The uptake of EVs by the cancer cells was also assessed by flow cytometry. PANC-1 cancer cells differing by the p53 status (CAS9 scrambled vs. CAS9-p53-KO) were simultaneously exposed to EVs from PANC-1 scramble and p53 KO cells. As observed in Figure 3E, there was no preferential uptake of mutant p53 harboring EVs over EVs that lacked mutant p53 protein. However, the overall efficiency of EV uptake was increased when the receiving cells had a GOF mutant p53. The double positive population was significantly higher in PANC-1 CAS9 scramble cells when compared to PANC-1 CAS9-p53-KO cells (Figure 3F), which indicates a potential advantage in uptake for the EV population with GOF p53 mutation. 

### 3.3. EVs Enriched with GOF Mutant p53 Proteins Can Modulate Macrophage Phenotypes

Macrophages constitute a significant portion of TME, as up to 50% of tumor mass can encompass macrophages [34]. We have previously reported that EVs from mutant p53 colorectal cancer cells reprogram macrophages [22]. We therefore aimed to delineate how the oncogenic mutant p53 protein shuttled via cancer-EVs will regulate macrophage signaling. To this end, we treated THP-1 macrophages with EVs from PANC-1 cells (CAS9 scrambled) and compared them with a treatment with EVs isolated from PANC-1 cells that lack mutant p53 (CAS9-p53-KO). As can be seen in Figure 4A, PANC-1 EVs were engulfed by the macrophages. In order to investigate the effect of EVs enriched with mutant p53 proteins on macrophage phenotype, we treated THP-1 macrophages with either PBS (control), PANC-1 p53 KO EVs, or PANC-1 CAS9 scramble EVs (EVs containing mutant p53 proteins). We measured the expression patterns of several pro-inflammatory or anti-inflammatory cytokines by quantitative real time PCR (qRT-PCR). Treating macrophages with PANC-1 scrambled EVs significantly induced the upregulation of tumor-supportive genes such as IL-1β, TNFα, IL-6, and MMP9 as presented in Figure 4B. As TNFα gene was significantly upregulated by the oncogenic mutant p53 harboring EVs, we next took the medium for TNFα ELISA for protein secretion. As seen in Figure 4C, incubation of macrophages with PANC-1 scrambled EVs significantly increased (*p* value < 0.05) the TNFα secretion relative to macrophages treated with PANC-1 p53 KO EVs. These results suggest the dominance of oncogenic mutant p53 protein harboring cancer-EVs over mutant p53 protein lacking cancer-EVs on macrophage reprogramming.

### 3.4. Non-Epithelial Staining of Mutant p53 in Clinical Samples

The typical immunohistochemical staining with antibodies against the human p53 proteins lack the specificity that permits the differentiating between WT and various mutants of p53 [35]. In this study, we used a mutant p53 antibody specifically for distinguishing between the GOF mutant in the R175H position and other forms of p53, including the WT and other mutants as published in [28]. Using the R175H antibody that also detects the mouse equivalent (R172H), we evaluated its ability to detect mutant p53 in heterozygote fibroblasts also expressing the WT allele (172/+). As shown in Figure 5A and Appendix A, the R175H-specific antibody was able to detect the R172H protein in the 172/+ cells but not WT p53 in fibroblasts expressing WT p53, demonstrating the specificity of this antibody in a multi-cellular environment and highlighting its ability to detect mutant p53 in cells expressing WT p53. We then used formalin-fixed-paraffin-embedded (FFPE) tumor tissues resected from colorectal cancer (CRC) patients for whom the TP53 status is known [22]. From this cohort, we chose eight tumors differing by their TP53 status as follows: WT p53 (*n* = 3), mutant p53 (R175H) (*n* = 3), and other p53 mutants (*n* = 2, R248W and H193Y). Immunohistochemistry using the antibody against the R175H mutant, revealed a highly specific positive staining in all three R175H samples while no staining occurred in the WT or the other mutant samples (Figure 5B,C). As could be observed, many areas of the R175H are negatively stained, supporting the notion of tumor clonal heterogeneity, where the mutant p53 clone is typically a late event in CRC. Notably, while some of the positively stained areas were in the epithelial compartment, we also found numerous non-epithelial compartments positive for mutant p53 staining and this supports our hypothesis of the potential protein transfer (Figure 5D). Pathological analysis indicated that adjacent to the cancer cells that were intensely positive for anti-R175H staining were immune cells that were also positively stained by the R175H antibody.

In order to further verify if mutant p53 can indeed be transferred out of the tumor mass into the TME, we utilized the human colorectal Colo-320DM cancer cell xenograft model, which expresses the R248W p53 mutant [36]. FFPE sections of subcutaneous tumors derived from the Colo-320DM xenografts were stained for p53 using the DO-1 antibody that specifically recognizes human p53 [35]. As expected, the tumors were stained positive by DO-1 (Figure 5E area marked “A”). However, we also noticed sporadic but specific staining outside the tumor mass in areas adjacent to the tumor (in areas marked “B” and “C”). In order to evaluate if these extra-tumoral staining were due to infiltrating tumor cells, we utilized another antibody recognizing a human nuclear protein (SRSF10). While the tumor mass was positively stained by the anti-SRSF10 antibody as expected, no specific staining was observed in the extra-tumoral cells and this indicates that the p53 staining in these areas are not due to tumor cell infiltration. These data collectively demonstrate that human p53 can be found in the TME adjacent to the tumor mass, both in human cancer contexts and in xenograft models, and this corroborates with our findings of mutant p53 being excluded via the EVs from the mutant p53-expressing tumors.

## 4. Discussion

Tumors prosper when the local tissue homeostasis is disturbed. In order to establish and maintain the supportive microenvironment, cancer cells have evolved mechanisms based on the recruitment of non-epithelial untransformed normal cells to support their sustained proliferation and their evasion of immune surveillance. The paper by Patocs and colleagues [26], which was published in 2007, challenged the entire accepted dogma regarding oncogenic mutations driving tumorigenesis in epithelial tumors. If, indeed, TP53 mutations also arise in the stromal cells thus granting those fibroblasts with oncogenic properties, many fundamental aspects of the TME have to be studied differently. While the sequencing data from this study have since been considered an artifact, the IHC staining that that article suggested that mutant p53 proteins are indeed found in the fibroblasts. This led us to conjecture that while no TP53 mutation occurs in the non-epithelial part of the tumor, the mutated p53 proteins do transfer exogenously. At the receiving cellular end, these GOF proteins might play an oncogenic role serving the needs of the cancer cells to promote progression and metastasis. While the vesicular transfer of mutant p53 between cancer cells and macrophages was presented in this report, the oncogenic potential of such delivery is yet to be determined, calling for additional research efforts.

Although a variety of signals that mediate the cross talk between cancer cells and their microenvironment has been identified, we still lack a general understanding of how the TME can be reprogrammed to promote the evolution of aggressive malignancy. Several types of cells from the TME have been shown to be subverted by cancer cells to support the latter’s progress through malignancy. For instance, naïve macrophages were shown to acquire features of tumor associated macrophages (TAMs) when co-cultured with cancer cells [37]. Similarly, untransformed fibroblasts become cancer-associated fibroblasts (CAFs) capable of supporting the growth of tumor cells [38]. While these characteristics of TME modulation are well established, the biochemical basis of TME conversion is not well understood.

Mutant p53 cancer cells were previously found to affect the size of the EVs, miRNA cargo and protein cargos of the EVs [39]. However, in our study, mutant p53 depletion did not affect the physical properties of EVs, which were in-line with our previous findings [22]. Furthermore, we did not observe any preferential uptake of EVs when they were loaded with GOF mutant p53 proteins (Figure 3E). Lack of discrepancy in the uptake of GOF mutant p53 protein harboring EVs over EVs lacking GOF mutant p53 proteins may occur due to several reasons: (i) Smaller EVs are known to be internalized more rapidly than the larger EVs [40]. However in our study, the size of EVs was not altered with the depletion of the mutant p53 protein, resulting in the non-selective uptake of EVs. (ii) It has been reported that recipient cells efficiently take up EVs from the same cell type/origin [41]. Our uptake experiments included EVs (with or without GOF mutant p53 protein) from PANC-1 cells added to the donor cells themselves, which might result in the non-differential uptake of EVs. On the other hand, as seen in Figure 3F, the presence of GOF mutant p53 protein in the recipient cells significantly improved the overall efficiency of EV uptake.

Ample evidence are showing that cancer-EVs can modulate macrophage reprogramming due to the ability of molecular cargos carried by cancer-derived EVs to educate macrophages switching to M1 type [42], M2 type [22,43], or to a mixed M1–M2 phenotype [44]. Alteration of tumor-supportive macrophage phenotypes renders the macrophages a status of tumor associated macrophages (TAMs). TAMs or educated macrophages by cancer-EVs imparts an aggressive trait to the underlying cancer [34]. Previously, we have reported the ability of mutant p53 cancers to reprogram macrophages in order to support tumor progression via EVs or exosomes [22]. Similarly, mutant p53 cancer was shown to transfer its invasive traits via EVs as evidenced by the alteration of the extracellular matrix microenvironment (ECM) [23]. However, the impact of oncogenic mutant p53 protein transferred via EVs by mutant p53 cancer cells still remains to be an avenue for deeper understanding. Notably, a recent study demonstrates oncogenic mutant p53 proteins being shuttled via EVs by cancer cells with a gain of function mutation to reprogram the fibroblasts into becoming cancer associated fibroblast (CAFs) [29]. Herein, we explored the EV based delivery of oncogenic mutant p53 proteins to another substantial member of TME, which are macrophages. Additionally, we show that oncogenic mutant p53 harboring EVs have a greater capacity to induce tumor-supportive cytokine production by macrophages, unlike EVs from cancer cells lacking mutant p53. The protein levels of TNFα were also increased in the recipient macrophages when incubated with oncogenic mutant p53 trapped EVs. Increased expression and secretion of TNFα and IL-6 as seen in our study is in line with other reports where breast cancer-EVs were shown to increase these cytokines [45,46]. Overall, our findings delineate a dominancy of oncogenic mutant p53 harboring EVs in macrophage reprogramming that can aid tumor advancement. Furthermore, the refractoriness of multiple cancer drugs can also be attributed to the presence of GOF mutant p53 protein, as supported by many studies. A study delineated that traditional cancer drugs such as 5-flurouracil, doxorubicin, and oxaliplatin were incapable of activating PUMA-mediated apoptosis in colorectal cancer cells, rendering patients with GOF mutation in p53 likely to be resistant to these drugs [47]. Notably, gemcitabine, a primary drug used to treat advanced PDAC was found to be less sensitive to cells with GOF mutant p53. However, gemcitabine in combination with small molecules (CP-31398 or RITA) having the ability to restore WT-p53 structure in the cells meliorated the responsiveness of gemcitabine [48]. The findings of our study on the transfer of GOF mutant p53 protein to non-transferred cells and neighboring cancer cells via EVs ferrets a speculation on the non-cell autonomous role of mutant p53 protein in priming the niche for cancer drug refractoriness.

In this context, the data presented here demonstrates the following: (i) mutant p53 protein can be selectively exported via EVs; (ii) mutant p53 in EVs can be taken up by neighboring cells in culture; finally (iii) mutant p53 expression is found in non-tumor cells in both human cancers and in non-human tissues in human xenografts. These data collectively highlight a novel mechanism by which mutant p53 can be transferred to other cells types and likely results in the exhibition of its GOF properties. The presence of mutant p53 in the TME is therefore expected to promote tumorigenesis and thereby establishes a positive feedback loop. Moreover, we suggest a novel GOF mechanism for mutant p53 as a master regulator of the TME (as depicted in Figure 5F). Further understanding the molecular basis of these observations and the signaling driven by mutant p53 expression in the TME will allow us to investigate the non-cell-autonomous functions of mutant p53 in the tumorigenic process.

## 5. Conclusions

In recent years, it has become clear that in addition to the classical cell-to-cell communication patterns, extracellular vesicles are instrumental in conveying molecular messages across cell types and have the potential to mediate changes at a tissue level. Since it is now appreciated that carcinomas are fundamentally reliant on two-way communication with activated cells in the TME, the elucidation of the role of EVs exchange and the role of the cargo that is transferred is essential to gain a thorough understanding of tumor progression. Establishing the non-cell autonomous role of mutant p53 in driving tumorigenesis has the potential to benefit future studies of the cross talk between the cancer cell and the TME. This will demonstrate the existence of an extra layer of communication that is orchestrated by small vesicles that transfer oncogenic elements between cellular entities. It will be intriguing to learn whether mutant p53 is unique in its microenvironmental activity or whether it represents a more general property of oncogenic stress responses. Building on the foundation of our proposed work on mutant p53, future studies may aim to characterize the potential activation of additional oncogenes, thus opening new paths of research at the interface of extracellular vesicles, cancer, and evolution.

## Figures and Tables

**Figure 1 cancers-13-02985-f001:**
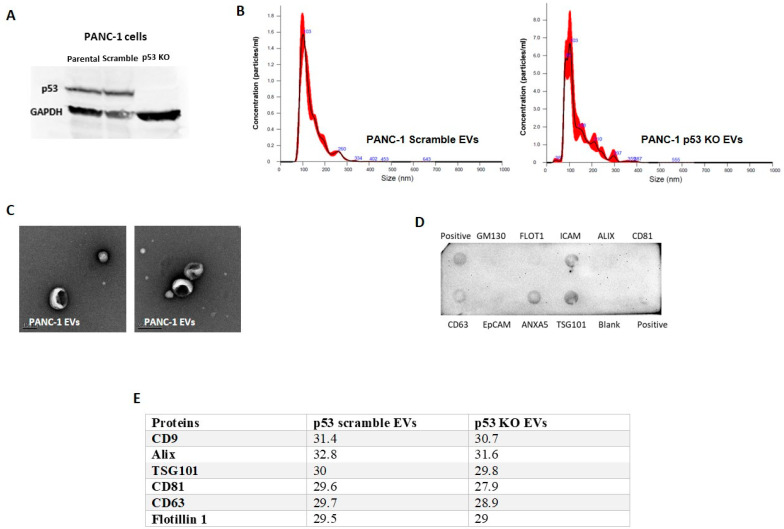
Characterization of EVs from mutp53-harboring cancer cells. The R273H GOF mutant p53 in PANC-1 cells was knocked out (using CRISPR Cas-9) as depicted in (**A**), where lysates from parental PANC-1 cells, Cas-9 scrambled, and Cas-9 KO were incubated with a p53 antibody (GAPDH was used as a loading control). EVs from PANC-1 cells were isolated using serial ultracentrifugations (UC) and were characterized using several methodologies including (**B**) nanoparticle tracking analysis (NTA), (**C**) electron microscopy, (**D**) exo-check membrane with several exosomal markers, and (**E**) proteomic comparison of several exosomal markers between scramble EVs and KO EVs. Log2(0) was replaced with 21 (the threshold intensity in the project).

**Figure 2 cancers-13-02985-f002:**
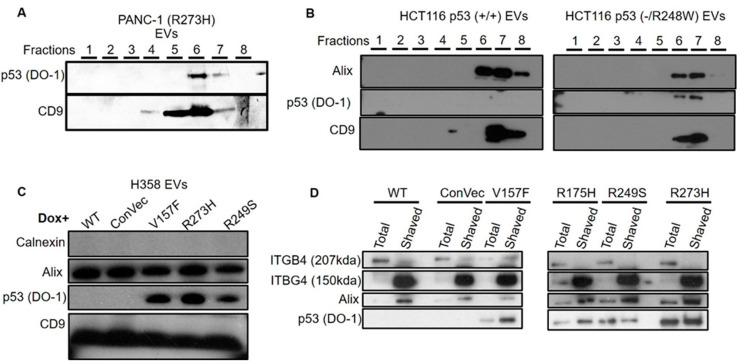
Mutant p53 proteins are sorted into EVs shed by cancer cells. EVs from several cancer cell lines harboring mutant p53 were isolated and their lysates were tested for the presence of the mutant p53 protein in WB. (**A**) Sucrose gradient (8 fractions) of PANC-1 cells harboring the p53 mutant R273H. (**B**) Sucrose gradient (8 fractions) of EVs from HCT116 cells expressing WT (+/+) p53 were compared to EVs from HCT116 cells harboring the mutant p53 (−/R248W). (**C**) EVs from H358 cells with Tet-On system expressing different p53 forms (ConVec = empty vector, no p53) CD9 and Alix were used as EV markers; calnexin was used as a cellular marker for confirming that no cellular contamination is present (**D**) EVs from H358 were treated with enzymatic proteolysis shaving of the external proteins. “Total”—no shaving treatment; “shaved”—with shaving treatment. ITBG4 was used as an external EV marker, while Alix was used as an internal EV marker.

**Figure 3 cancers-13-02985-f003:**
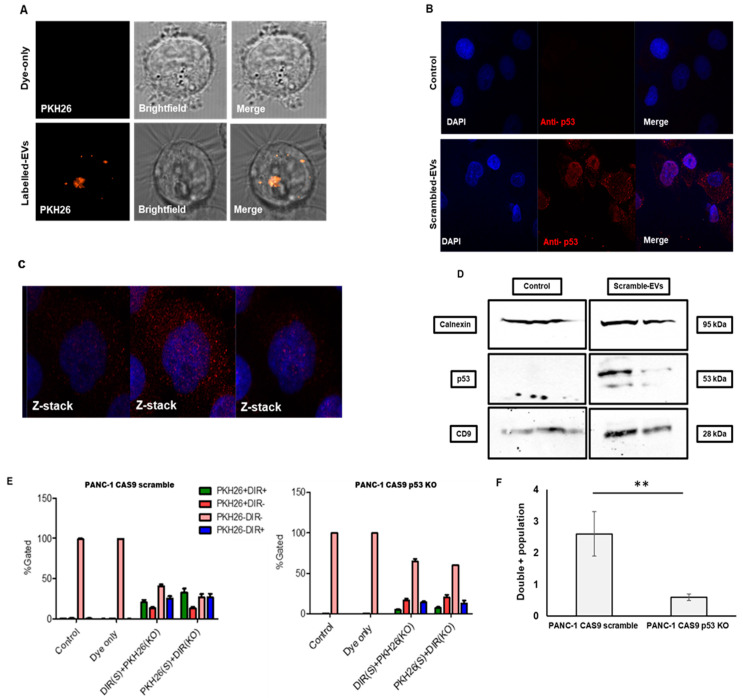
Mutant p53 from PANC-1 EVs are taken up by PANC-1 p53 KO cells. (**A**) EVs from PANC-1 scrambled cells were isolated, labelled with PKH26 dye, and added to PANC-1 p53 KO cells for 18 h before being scanned in a laser scanning confocal microscope (63X, Zeiss Plan-Apochomat oil, 1.4 NA). Orange—PKH-26 labelled PANC-1 EVs. (**B**) Non-labelled EVs were added to PANC-1 p53 KO cells for 18 h and immunoflurescence microscopy (63X) was utilized to reveal mutant p53 protein. (**C**) Z stack of the IFM capture. (**D**) Immunoblot of PANC-1 p53 KO treated with EVs from PANC-1 scrambled for 24 h (duplicated lanes). Calnexin served as the cellular marker and CD9 as the vesicular marker. (**E**) EVs were labelled with PKH26 and DIR- and simultaneously exposed to PANC-1 cells (CAS9 scrambled and CAS9 p53 KO). (**F**) Bar graph showing an increase in EV uptake in the presence of GOF mutant p53 protein in the recipient cells. Uptake efficiency was measured via flow cytometry. Briefly, 15,000 events were noted and analyzed via cytexpert software. The experiment was repeated at least three times (*n* = 4). A two tailed Student’s *t*-test was performed to determine statistical significance ** <0.005.

**Figure 4 cancers-13-02985-f004:**
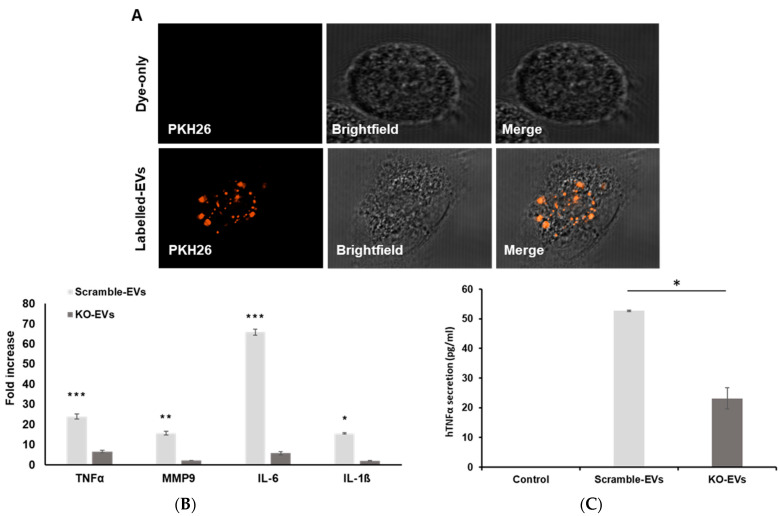
EVs enriched with GOF mutant p53 proteins can modulate macrophage phenotypes. THP-1 derived macrophages were treated with either (**A**) dye only (control) or PKH26-labelled PANC-1 scrambled EVs and visualized using confocal microscopy (63X, Zeiss Plan-Apochomat oil, 1.4 NA). (**B**) PBS (control), PANC-1 scramble EVs or PANC-1 p53 KO EVs, and qRT-PCR were performed for different genes relative to a housekeeping gene (GAPDH). The fold change in gene expression is relative to control groups (normalized to 1 in the graph). (**C**) PBS (control), PANC-1 scramble EVs or PANC-1 p53 KO EVs, and medium were subjected to hTNFα ELISA. A representative result of three biologic repeats is shown. A two tailed Student’s *t*-test was performed to determine statistical significance * <0.05, ** < 0.005, and *** <0.0005.

**Figure 5 cancers-13-02985-f005:**
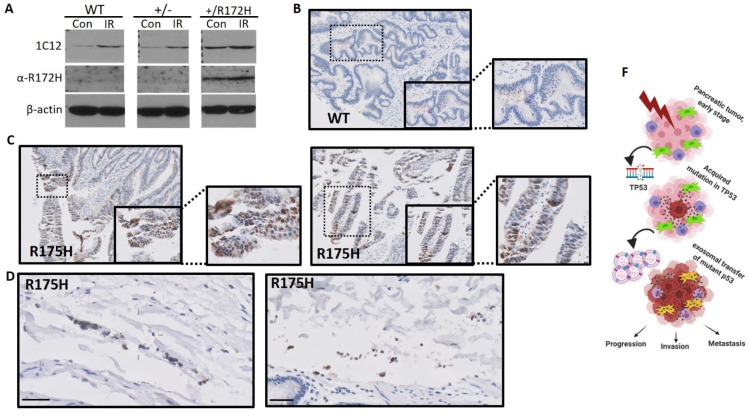
Non-epithelial staining of mutant p53 in clinical samples. Mutant p53-specific immunohistochemistry of resected tumors from CRC patients were sequenced and the TP53 status was determined. Eight cases were stained using the anti-R175H antibody. (**A**) Mouse tail-tip fibroblasts expressing wild-type (WT) p53, heterozygous for p53 (+/−), and heterozygous for mutant R172H p53 (172/+) were treated without (Ctrl) or with 10 Gy g-irradiation (IR) and harvested 2 h later and used for immunoblot analysis. IC12 is a pan-p53 antibody that detects all forms of p53. The a-R172H antibody is specific for the R172H mutation (human R175H). Loading was controlled with b-actin antibody. (**B**) Representative WT case with no R175H (anti-R175H-p53) staining detected. (**C**) Two representative mutant p53 R175H images in which specific nuclear positive R175H staining is detected in tumor nests as well as in areas outside the tumor nests. (**D**) Non-epithelial areas in the R175H cases where nuclear staining is detected; bars = 50 μm. (**E**) Colo-320DM human colorectal cells were used to generate xenograft tumors, which were harvested and stained with DO-1 antibody to detect mutant p53 expression or the SRSF10 antibody to mark cells of human origin. Representative images are shown. White arrows are for orientating the images and area “A” marks the tumor core tissue. Areas “B” and “C” show the presence of mutant p53 stains in non-tumor tissues as they are not stained by the SRSF10 antibody. Negative control images of tumors stained without primary antibodies are shown. Images denoted with 20× and 40× represent a 200× and 400× magnification respectively. Images of staining without primary antibodies are 100× magnified (**F**) Suggested model: Mutations in p53 are a common event in many cancer types. When a tumor cell acquires a GOF mutation in p53, it will release EVs enriched with the mutant p53 protein. Such EVs are taken up by neighboring tumor cells, macrophages, fibroblasts, and other tumor microenvironment (TME) members. Expression of mutant p53 in target cells will result in oncogenic mechanisms in the TME, which drives tumor progression and eventually metastasis.

**Table 1 cancers-13-02985-t001:** List of primers.

Genes	Forward Primer	Reverse Primer
TNFα	CACTTTGGAGTGATCGGCCC	AGCTTGAGGGTTTGCTACAAC
IL-6	CACTCACCTCTTCAGAACGAAT	GCTGCTTTCACACATGTTACTC
IL-1β	CCTTAGGGTAGTGCTAAGAGGA	AAGTGAGTAGGAGAGGTGAGAG
MMP9	GGCACCACCACAACATCACC	GATACCCGTCTCCGTGCTCC

## Data Availability

The data presented in this study are available upon request from the corresponding author.

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
