# Peer review of "Cancer Cells Shuttle Extracellular Vesicles Containing Oncogenic Mutant p53 Proteins to the Tumor Microenvironment"

_cancers, 2021, doi:10.3390/cancers13122985_

Round 1
Reviewer 1 Report
In the current research study, the authors found mutant p53 proteins in EVs can be transferred to macrophages and modulate the release of tumor supportive cytokines. The authors claim that specific gain-of-functions (GOF) p53 mutants can promote oncogenic mechanisms via reprogramming and recruiting non-cancerous elements for tumor progression. The authors need to perform new experiments to address further questions listed below. Overall, this manuscript needs major revision.
- The order of Figures 1C and 1D is not consistent with the description in the result part. Figure 1 legend “(C) exo-check membrane with” should be “(D) exo-check membrane with”. Figure 1D needs to be further explained, such as which are exosome markers, what are positives.
- It would be great if the authors can determine whether P53 is located on the membrane of EVs. A Proteinase K protection assay and ImmunoTEM assay could be used.
- Statistical methods and sample size should be included in the figure legends for figures 3E-F and 4B-C.
- Figure 4B, it is not clear what the two bars represent. It is not clear how many/much EVs were used to treat the cells.
- Details of the human tissues used should be included.
Author Response
In the current research study, the authors found mutant p53 proteins in EVs can be transferred to macrophages and modulate the release of tumor supportive cytokines. The authors claim that specific gain-of-functions (GOF) p53 mutants can promote oncogenic mechanisms via reprogramming and recruiting non-cancerous elements for tumor progression. The authors need to perform new experiments to address further questions listed below. Overall, this manuscript needs major revision.
We wish to thank the reviewer for the time invested in reviewing our manuscript and for the thoughtful and insightful comments! We have addressed all comments as detailed below (in bold).
- The order of Figures 1C and 1D is not consistent with the description in the result part. Figure 1 legend “(C) exo-check membrane with” should be “(D) exo-check membrane with”. Figure 1D needs to be further explained, such as which are exosome markers, what are positives.
We have changed the order of the legends as suggested and added further details about the kit in the methods section as well as in the legend itself.
- It would be great if the authors can determine whether P53 is located on the membrane of EVs. A Proteinase K protection assay and ImmunoTEM assay could be used.
We have conducted an “exosome shaving” experiment where we concluded that mutant p53 is located inside the vesicles. In this experiment, we used an enzymatic digestion of proteins located outside the EVs or on their surface. Proteins inside the EVs were protected from this digestion. As could be seen in Figure 2, we used two protein markers: one for the surface (ITGB4) and one for the inner part of the EVs (Alix). Since mutant p53 in these EVs behaved like Alix and was protected from the shaving, we concluded that indeed the mutant p53 is inside the EVs and not on the surface.
- Statistical methods and sample size should be included in the figure legends for figures 3E-F and 4B-C.
We have included the statistical methods and sample sizes in the requested figures.
- Figure 4B, it is not clear what the two bars represent. It is not clear how many/much EVs were used to treat the cells.
These requested details are now included. A clear legend has been added to the figure 4B. Three doses of 20µg of EVs, every 12 hours, was added to the macrophages. It is described in the method section under subheading macrophage generation and subculture, and now also added in the legend.
- Details of the human tissues used should be included.
These requested details are now included.
Reviewer 2 Report
This is an interesting manuscript which provides important evidence on gain of function mutant p53 proteins as mediators of oncogenic activities to surrounding cells.
The evidence that mutant p53 proteins can be detected in EVs is of great relevance and sheds light on how mutant p53 proteins might contribute to instruct aberrantly cells surrounding tumoral mass.
The discussion section should be more speculative also implying the potential role that EVs containing mutant p53 proteins might have in chemoresistance to conventional and precision anticancer drugs.
Based on this the manuscript warrants acceptance for publication in Cancers upon minor revision.
Author Response
This is an interesting manuscript which provides important evidence on gain of function mutant p53 proteins as mediators of oncogenic activities to surrounding cells.
The evidence that mutant p53 proteins can be detected in EVs is of great relevance and sheds light on how mutant p53 proteins might contribute to instruct aberrantly cells surrounding tumoral mass.
We wish to thank the reviewer for the time invested in reviewing our manuscript and for the thoughtful and insightful comments! We have addressed all comments as detailed below (in bold).
The discussion section should be more speculative also implying the potential role that EVs containing mutant p53 proteins might have in chemoresistance to conventional and precision anticancer drugs.
We included an additional paragraph discussing chemoresistance in the context of mutp53 and its potential exosomal delivery.
Based on this the manuscript warrants acceptance for publication in Cancers upon minor revision.
Reviewer 3 Report
This manuscript presents some interesting findings. However, there are several issues which must be addressed before this paper can be considered for publication.
Interpretation of the data presented is made significantly more challenging by the fact that in multiple places the data shown in the panels of figures does not match up with the description in the figure legends, or in places in the main text.
In Figure 1, the authors start in panel A by comparing 4 different cell lines, the next panel shows 3 of the cell lines, then next panel only one of the cell lines and then the last two panels show data for 2 of the cell lines. Where is the rest of the data? It is very hard to understand or determine if the conclusions the authors are drawing are accurate with a significant amount of data missing from the figure.
Some of the western blots are currently not publication quality and need to be replaced prior to publication - in particular CD9 in Figure 2C. Again in the description of the western blots in figure 2 - the proteins which shown in the panels, don't always match up with the description in the legend. Some of the images are also too closely cropped to the bands, the bands need to be in the middle of the section of the blot that is being presented. Additionally, as the authors are comparing cell lines with and without p53 it would be helpful if a positive control is present on every blot so the blots of p53 KO can be confidently interpreted.
As the authors have used multiple techniques and cell lines, the descriptions of the samples being present in the figures need to match with the description of the data in the main text. Currently it is very hard to determine which sample is which as there is a lack of consistency between text and labels in the figures.
Examination of the data suggests that the authors indeed have some interesting and potentially exciting findings, but in its current state this manuscript is difficult to follow and therefore cannot be evaluated scientifically. Furthermore, it has too many errors for publication. I would recommend that this manuscript is reconsidered once, better quality western blots are provided with additional positive controls for p53 and the authors have corrected the errors relating to the presentation and description of the data.
Author Response
This manuscript presents some interesting findings. However, there are several issues which must be addressed before this paper can be considered for publication.
We wish to thank the reviewer for the time invested in reviewing our manuscript and for the thoughtful and insightful comments! We have addressed all comments as detailed below (in bold).
Interpretation of the data presented is made significantly more challenging by the fact that in multiple places the data shown in the panels of figures does not match up with the description in the figure legends, or in places in the main text.
We are extremely sorry for any inconvenience in reading the manuscript, we made an effort to clarify and present the findings in a better way.
In Figure 1, the authors start in panel A by comparing 4 different cell lines, the next panel shows 3 of the cell lines, then next panel only one of the cell lines and then the last two panels show data for 2 of the cell lines. Where is the rest of the data? It is very hard to understand or determine if the conclusions the authors are drawing are accurate with a significant amount of data missing from the figure.
We now present only the CRISPR-CAS9 knock-out blots and NTA results which are used in the following sections of the manuscript.
Some of the western blots are currently not publication quality and need to be replaced prior to publication - in particular CD9 in Figure 2C. Again in the description of the western blots in figure 2 - the proteins which shown in the panels, don't always match up with the description in the legend. Some of the images are also too closely cropped to the bands, the bands need to be in the middle of the section of the blot that is being presented. Additionally, as the authors are comparing cell lines with and without p53 it would be helpful if a positive control is present on every blot so the blots of p53 KO can be confidently interpreted.
We modified figure 2C and improved the quality of the presented CD9. We also reformed the legend of figure 2 to coincide with the presented data. We enlarged the cropped area of some of the bands to position them in the middle of the cropped section. As for the positive p53 control, we included the comparison between mutant p53 and the KO in several cells and EVs. Please note the following figures: 1A (PANC1 cells with and without KO), 2C, 2D (EVs from H358 cells, ConVec samples have no p53 while WT have the WT form and the rest are mutant forms of the protein), 5A (WT and mutant forms of p53 from different genetic models of mice).
As the authors have used multiple techniques and cell lines, the descriptions of the samples being present in the figures need to match with the description of the data in the main text. Currently it is very hard to determine which sample is which as there is a lack of consistency between text and labels in the figures.
We meticulously went over all legends and panels to ensure clarity. As could be seen in ‘track-changes’ in the legends and the figures, we made significant modification to assure such clarity and add details.
Examination of the data suggests that the authors indeed have some interesting and potentially exciting findings, but in its current state this manuscript is difficult to follow and therefore cannot be evaluated scientifically. Furthermore, it has too many errors for publication. I would recommend that this manuscript is reconsidered once, better quality western blots are provided with additional positive controls for p53 and the authors have corrected the errors relating to the presentation and description of the data.
Round 2
Reviewer 3 Report
I thank the authors for thoroughly checking the revised manuscript and correcting the spelling and inconsistencies with the labelling of panels in the figures and their description in the text.
My only remaining concern is in the legend for Figure 4 the authors have stated for panel C, the data shown is for 2 repeats. They have then shown error bars and statistical analysis of this data, however n=2 does not represent sufficient data points for these tests to be meaningful or robust. Please could the authors clarify how many data points have been included in this analysis.
Author Response
We thank the reviewer for this comment!
We can confirm 3 independent biological repeats. The duplicates are technical. We changed the wording of the figure so there would be no misinterpretation.